# Phyto-Inhibitory and Antimicrobial Activity of Brown Propolis from Romania

**DOI:** 10.3390/antibiotics12061015

**Published:** 2023-06-05

**Authors:** Mihaela Laura Vică, Mirel Glevitzky, Ramona Cristina Heghedűş-Mîndru, Gabriela-Alina Dumitrel, Gabriel Heghedűş-Mîndru, Maria Popa, Doriana Maria Faur, Ștefana Bâlici, Cosmin Adrian Teodoru

**Affiliations:** 1Department of Cellular and Molecular Biology, “Iuliu Hațieganu” University of Medicine and Pharmacy, 400012 Cluj-Napoca, Romania; mvica@umfcluj.ro (M.L.V.); sbalici@umfcluj.ro (Ș.B.); 2Institute of Legal Medicine, 400006 Cluj-Napoca, Romania; 3Faculty of Exact Science and Engineering, “1 Decembrie 1918” University of Alba Iulia, 510009 Alba Iulia, Romania; mirel_glevitzky@yahoo.com (M.G.); mpopa@uab.ro (M.P.); 4Faculty of Food Engineering, University of Life Science “King Mihai I”, 300645 Timișoara, Romania; ramonaheghedus@usab-tm.ro (R.C.H.-M.); gabrielheghedus@usab-tm.ro (G.H.-M.); 5Faculty of Industrial Chemistry and Environmental Engineering, Politehnica University of Timisoara, 300223 Timișoara, Romania; 6Clinical Emergency Hospital for Children, 400398 Cluj Napoca, Romania; popa.dorianaa@yahoo.com; 7Clinical Surgical Department, Faculty of Medicine, “Lucian Blaga” University of Sibiu, 550025 Sibiu, Romania; adrian.teodoru@ulbsibiu.ro

**Keywords:** propolis, chemical analysis, antibacterial, antifungal, phyto-inhibitory activity, statistics

## Abstract

The objective of this paper was to study the phyto-inhibitory and antimicrobial activity of brown propolis collected from the counties of four regions in Romania. The main physico-chemical and functional properties of 16 samples of propolis from different landforms of geographical regions were determined. Their antimicrobial activities were established against 5 bacterial strains (*Pseudomonas fluorescens*, *Bacillus subtilis*, *Bacillus cereus*, *Escherichia coli*, and *Proteus mirabilis*) and 5 fungal strains (*Alternaria alternata*, *Cladosporium cladosporioides*, *Fusarium oxysporum*, *Mucor racemosus*, and *Aspergillus niger*). Simultaneously, the phyto-inhibitory effect of propolis samples on different cereals was highlighted: hexaploid bread wheat (*Triticum aestivum*), maize (*Zea mays* L.), oats (*Avena sativa* L.), and barley (*Hordeum vulgare* L.). Correlations between the antioxidant activity and total flavonoid and phenol content of the propolis samples were identified, respectively, and the statistical analysis highlighted that the diameter of the inhibition zone was influenced by the strain type (bacterial and fungal) and the geographical regions of propolis. Principal component analysis (PCA) indicated that out of seven principal components, only two exhibited > 0.5. Pearson’s correlation coefficient showed a low and moderate positive linear relationship between the diameter of the inhibition zone and the flavonoid and phenol concentration of the propolis samples.

## 1. Introduction

Propolis (bee glue) has been known and used since ancient times in traditional folk medicine [1]. Different biological properties have been attributed to propolis as a preservative, as a potentially functional product, as an antioxidant, with growth-inhibitory effects, antimicrobial, anti-inflammatory, antiulcer, wound healing, antitumor, anti-angiogenin, anti-hyperlipidemic, and immunomodulatory activities [2,3,4,5,6,7,8,9,10].

Propolis is a natural resinous product collected by honeybees (*Apis mellifera* L.) from various plants and then mixed with salivary and enzymatic secretions [11]. It is composed of resin, wax essential oils, pollen, and organic compounds [12]. In the propolis, more than 500 compounds have been identified and classified as follows: 50–70% resin and balsams, 30% wax, 10% essential oils, 5–10% pollen, and 5% other constituents, containing sugars, minerals, vitamins B, C, and E, cinnamic acid, flavonoids, phenolic acids, and their esters [13,14,15]. Moreover, other chemical components are aldehydes, aromatic compounds, alcohols and ketones, steroids, coumarins, amino acids, and inorganic compounds [16,17].

The effects and chemical compounds of propolis depend on its type, geographical origin, plant sources, year, and time of sampling [18]. The composition of propolis is important when it is used in the food industry, cosmetics, and pharmaceutical industries [19].

The antioxidant and antimicrobial activity of propolis have been studied so far by using alcoholic or aqueous extracts of this bee product [20]. Phenolic compounds from propolis are responsible for antioxidant activity. The neutralization of free radicals is generated also by antioxidants such as vitamin C, flavonoids, and polyphenols from propolis. The role of antioxidants from propolis is to neutralize the excess of free radicals with the formation of more stable molecules [21]. In addition, flavonoids and phenolic acids have good antimicrobial activity [8,22,23,24,25].

Ever since 1965, a real “fever” of patents has been observed regarding the use of propolis. Its use in agriculture, among other fields, was not omitted [26]. Propolis in agriculture is mainly used to combat phytopathogens in crops. The effect of ethanolic propolis extract on anthracnose severity, growth, and productivity of Carioca berries was investigated [27], but there are studies on the use of propolis in the cultivation of tomatoes (to control tomato bacterial wilt caused by *Ralstonia solanacearum*), coffee (to control coffee leaf rust), cucumbers (for the control of powdery mildew by preventive application of a concentration of 8%.), and grapes (for controlling the growth of *Aspergillus flavus*) [28,29,30,31,32,33]. The in vitro and in vivo antifungal activity of Brazilian red propolis against *Colletotrichum musae* was also studied as a potential natural alternative for the control of banana crown rot [34].

Propolis extracts have also been studied for foliar application or soil irrigation on fava bean plants and their role against nematode infection [35]. The antifungal potential of propolis alcoholic extracts was investigated on the mycelial growth of *Botrytis cinerea* which affects many species of plants, fruits, and vegetables [36].

Taking all these into account, the aim of our study was to determine the physico-chemical characteristics of several brown propolis samples collected from four regions of Romania (Banat, Crișana, Maramureș, and Transylvania), as well as to evaluate the antimicrobial activity of these samples against various bacterial and fungal strains that contaminate cereals and to determine the phyto-inhibitory activity of propolis on some cereal species. At the same time, correlations between the antioxidant activity and total flavonoid content of the propolis samples were investigated. In addition, we investigated whether the antimicrobial activity of the propolis samples was influenced by the type of strain and the geographical regions of origin.

## 2. Results

### 2.1. Physico-Chemical Analysis

At the beginning of the study, the brown propolis samples collected from different regions of Romania (Banat, Crișana, Maramureș, and Transylvania) were characterized from a physico-chemical point of view. The values of water activity, water solubility, total phenols, total flavonoids, and the FRAP and DPPH are presented in Table 1.

Brown propolis samples from different geographical locations of the Banat, Crisana, Maramureș, and Transylvania regions recorded a water activity range from 0.62 to 0.74, where only a few microorganisms can still grow. Propolis water solubility varies between 8.74 and 15.61%, justifying the fact that it is poorly soluble in water.

The total phenolic content of the propolis samples ranged between 108.2 and 193.4 mg GAE/g, while the flavonoid content was 53.72 and 97.65 mg QE/g. The highest content of phenols was found in sample S5 (193.4 mg GAE/g) and flavonoids in sample S14 (97.65 mg QE/g), respectively.

The antioxidant capacity values were obtained using two methods: by ferric-reducing antioxidant power (FRAP) and 2,2-diphenyl-1-picrylhydrazyl (DPPH) radical scavenging record values between 0.25 and 2.14 mmol Fe^2+^/g and 11.2 and 18.11 mg GAE/g, respectively.

### 2.2. The Phyto-Inhibitory Activity of Propolis

Table 2 shows the average length of plume growth for 13 days in cereal samples: wheat, maize, barley, and oats treated with different concentrations of aqueous propolis solutions (APE) for all 16 counties.

In the case of the seeds, the radicle and the plumule, respectively, are clearly visible to the naked eye from the first days, especially in the case of the control samples. The slowing down of plumule growth increases with increasing propolis concentration over time.

In the case of the samples treated with 1% propolis solution, the highest growth of the plume compared to the control is in the case of wheat: 150 mm/145 mm = 1.034. The lowest length growth–control ratio is 81 mm/123 mm = 0.658 for oat. The highest length growth–control ratio for samples treated with 5% propolis solution was for wheat: 137 mm/145 mm = 0.945, and the lowest ratio (62 mm/123 mm = 0.504) for oat. For the 10% propolis solution, the highest length–control ratio was 83 mm/132 mm = 0.629 for barley, and the lowest ratio was 30 mm/83 mm = 0.361 for barley.

Figure 1a–d show the situation of cereal samples: wheat, maize, barley, and oats treated with different concentrations of APE after 13 days.

A difference in plume growth is observed between the samples treated with different APE concentrations at the end of the 13 days of monitoring for all studied grain samples.

### 2.3. Antimicrobial Activity of Propolis

Table 3 shows the results of the antibacterial effect of the 16 propolis samples against the studied strains and the inhibition diameter area for a synthesis of antibiotic–ciprofloxacin, respectively.

All propolis samples showed antibacterial activity against all types of bacterial strains. The diameters of the inhibition zones ranged from 18 to 32 mm.

Table 4 shows the results of the antifungal effect of the propolis samples.

All propolis samples showed antifungal activity against all types of fungal strains. The diameters of the inhibition zones are smaller than in the case of bacteria, ranging from 15 to 27 mm.

### 2.4. Statistical Analysis

Figure 2 shows the correlation between the antioxidant capacity determined by the DPPH and FRAP methods and the total flavonoid content of the propolis samples.

Figure 3 shows the correlation between the antioxidant capacity determined by the DPPH and FRAP methods and the total phenolic content of the propolis samples.

A first-order multiple linear regression model was used to estimate the relationship between the grains’ plume growth (y) as a dependent variable, and two independent variables: time (x_1_) and the concentration of propolis extract applied as an inhibitor (x_2_). Variance (σ^2^), standard deviation (σ), correlation coefficient (r), and squared correlation coefficients (r^2^) were used as indicators of model adequacy. The equations and the concordance indicators of the determined statistical model are presented in Table 5.

Principal component analysis (PCA) was performed. The concentrations of considering APE—C(1%), C(5%), and C(10%), together with the control sample, were used as input data. The analysis was designed to evaluate the phyto-inhibitory effect of the three aqueous propolis solutions on the four cereal samples after 3, 5, 7, 9, 11, and 13 days.

Principal components (PCs) were determined from the eigenvalues of the correlation matrix of observations. The eigenvalues were found to be 3.81, 0.14, 0.035, and 0.005, for PC1 to PC4. As can be seen in Figure 4, the first two PCs explain 98.97% of the total variance. PC1 explains 95.29% and PC2 explains 3.68%, respectively.

Figure 5 shows the observations and PCs obtained from the data analyzed. The formation of two groups of cereal samples was observed. The first blue group, located on the upper right of the score graph, consists of the oat and barley samples after 9, 11, and 13 days of treatment with propolis solution. The second group marked in green, located at the bottom right of the score graph, consists of the maize and wheat samples after 9, 11, and 13 days of treatment with propolis solution. The grain samples marked in red, located in the center-left part of the score graph, showed a lower response to the propolis solutions during the first 3–7 days. Consequently, it can be stated that C(1%), C(5%), and C(10%) values (Figure 6) are useful in clustering the cereal groups. For the cereal group consisting of oat and barley samples after 9, 11, and 13 days, the variable C(10%) was responsible for the classification. For the cereal group consisting of maize and wheat samples after 9, 11, and 13 days, the variables responsible for the classification were C(1%) and C(5%).

The contribution of the variables to PCs is shown in Figure 6. The C(1%) and C(5%) concentrations of the propolis solutions strongly correlate with PC1 in the positive direction, while the C(10%) concentration of the propolis solution is strongly correlated with PC2 also in the positive direction. Consequently, it can be stated that the use of C(1%) and C(5%) propolis solutions led to good results regarding the phyto-inhibitory effect of maize and wheat samples after a period of 9 to 11 days, even from 7 days from wheat. The use of C(10%) propolis solution led to good results regarding the phyto-inhibitory effect of barley and oat samples after a period of 9–11 days.

A two-way ANOVA test analyzes the simultaneous effect of two independent variables: propolis extracts from Banat, Crișana, Maramureș, and Transylvania regions, Romania, and the diameter of the inhibition area for the studied strains (bacteria and fungi).

For the 16 samples of brown propolis, the dispersion caused by each variable parameter will result, including residual dispersion caused by accidental factors (Table 6).

Since F_computed_ for bacteria and fungi are both greater than F_0.05_, the null hypothesis that the mean values of the columns and rows are equal was rejected. It was concluded that the geographical origin of Romanian propolis and the type of strain (bacteria and fungi) influenced the diameter of the inhibition zone at a significance level α = 0.05.

In Table 7, the values of Pearson’s correlation coefficient show the strength and direction of a linear association between the diameter of the inhibition zone and the flavonoid and phenol content of propolis for all the microbial strains.

## 3. Discussion

Among all the bee products extracted from the bee colony, the most pronounced phyto-inhibitory activity is in the case of honey and propolis [37].

The properties and the content of propolis differ considerably from region to region along with vegetation, from season to season, and from hive to hive [38]. Its composition varies depending on geographic location, botanical origin, and climatic factors [39].

The physico-chemical parameters showed significant differences in propolis samples from the different counties of Transylvania. The water activity of the propolis samples varies between 0.62–0.74 and is not a proper environment for the majority of microorganisms. The values are similar to those of the studies carried out by Devequi-Nunes et al. [19]. The propolis samples displayed low water solubility (between 8.74 and 15.61%) according to the value obtained by Pant et al. [40]. Biologically active substances have low solubility in water, and the number of phenolic compounds in aqueous extracts is lower than in ethanolic extracts [41,42].

Some plant materials are being used now to mitigate the devastating effects of abiotic stresses on plants. Propolis also protects plants against viruses, bacteria, molds, and various fungi. Propolis contains essential compounds (flavones, sugars, aromatic acids, amino acids, vitamins, minerals, terpenes, and sesquiterpenes) with an impact on the activity of various physiological processes in plants [43].

The phyto-inhibitory and phytotoxic activity of propolis extracts is shown in many studies. For example, the potato tubers kept in the hive did not sprout and after being kept in the hive for a long time they suffered permanent inhibition. Moreover, the aqueous extract of propolis was found to be responsible for inhibiting the germination and growth of lettuce seedlings and rice grains [37]. Derevici et al. compared the inhibitory effect of aqueous propolis solution from Romania and Russia on the germination of certain seeds and revealed an inhibitory effect on hemp seeds (*Cannabis sativa*), which occurred at a dilution of 1:10 [44]. The presoaking application of propolis and maize grain extracts alleviates salinity stress in common beans (*Phaseolus vulgaris* L.) [45]. In finding new natural products with potential herbicide activity, propolis was studied by King-Díaz et al. [46] who isolated flavonoid compounds from propolis and tested them on the germination of *Lolium perenne*, *Echinochloa crus-galli,* and *Physalis ixocarpa* seedlings.

Analyzing the results of our tests regarding the phyto-inhibitory activity of the studied APE, it can be observed that the cereal species behaved similarly during growth, the phyto-inhibitory activity of propolis being evident in all four cereal species (Table 2, Figure 1). It can be also noticed that there is a similarity in terms of growth behavior under the conditions of administration of phyto-inhibitory agents for the wheat and maize samples for the barley and oat samples, respectively. The reduction in growth speed was significant in the case of the administration of larger amounts of APE (10% solution). For wheat, this reduction was almost 3 times, and for maize and oats 2–2.5 times. The reduction was less significant in the case of barley.

Our findings are in accordance with [47] who concluded that the propolis sample inhibits the germination of wheat seedlings (*Triticum durum* Desf. cv.Kunduru) with the increase of its concentration.

The results obtained are in accordance with the research of Dadgostar and Nozari [48] who tested propolis extracts on the germination of wild barley, oat, and cattle cotton.

Regarding the antimicrobial activity, our study showed that all propolis extracts had both antibacterial and antifungal effects on all of the analyzed strains, with some samples having a more pronounced antibacterial effect even than that of the antibiotic used as a control, in the case of some strains (*P. fluorescens, E. coli, P. mirabilis*).

Our results confirm the antimicrobial effect of propolis, demonstrated in other studies. Since the chemical composition of propolis varies considerably depending on the geographical area, the antibacterial activity of propolis is determined by its origin, the bee species, and the procedure used to obtain the extract [8]. In the case of bacteria, poplar propolis has been reported to have antibacterial effects against both Gram-positive and Gram-negative bacteria [49]. Numerous other studies have found that different types of propolis have significant antimicrobial activity against a wide range of bacterial pathogens, including those analyzed in this study [50,51,52]. In addition, our study confirms our previous results on the effect of some propolis samples on fungal strains [25], as well as other studies on *Alternaria* spp., *Fusarium* spp., *Mucor* spp., and *Aspergillus* spp. [53,54].

Considering that the analyzed propolis samples had a good antimicrobial effect against strains of bacteria and fungi that are frequently found on cereals, this can be a premise for the use of propolis as an antibacterial agent for such crops. On the other hand, since the phyto-inhibitory activity of these propolis samples has been demonstrated, the use of such products must be conducted with caution in order to not affect plant growth.

The differences regarding the bioactivity of propolis are due to the chemical variations of its constituents [55,56].

A strong correlation (R^2^ = 0.74) is observed between the antioxidant activity (DPPH) and flavonoid content for propolis samples from the four regions of Romania (Banat, Crișana, Maramureș, and Transylvania) (Figure 2). The FRAP assay also showed a moderate correlation between the two parameters with a correlation coefficient whose magnitude is 0.67. This confirms that flavonoids have antioxidant activity, and almost every group of flavonoids has the ability to act as antioxidants [57]. 

The correlation coefficients describe the strength of an association between the antioxidant capacity of propolis samples and their content in phenolic compounds (Figure 3). The results indicate a strong linear correlation between the RSA of 2,2-diphenyl-1-picrylhydrazyl and the phenolic compounds. The correlation coefficient (R^2^ = 0.69) shows a positive moderate correlation between total phenolic compounds and the total antioxidant capacity measured by ferric-reducing antioxidant power (FRAP).

The use of PCA on the growth lengths of plumes for the studied cereals is suitable for data reduction and summarization purposes. The first few principal component coordinates that explain most of the variance can be used as variables in further statistical analysis in exactly the same way as in standard principal component analysis (Figure 5 and Figure 6). The principal component analysis allowed the information content of large data tables to be summarized through a smaller set of “summary indices” that can be more easily visualized and analyzed.

The Pearson’s correlation coefficients between the diameter of the inhibition zone and the flavonoid and phenol content (Table 7) indicated that the coefficients 0.21 and 0.05 show negligible correlations in the case of *M. racemosus* and low and moderate in the rest of the cases, indicating a positive linear relationship between the two variables. The correlation is statically significant in the case of *E. coli* (Pearson’s correlation coefficient is 0.68 for flavonoids and 0.54 for phenols) and *A. niger* (Pearson’s correlation is 0.64 for flavonoids and 0.66 for phenols) indicating a better moderate positive linear relationship. On the whole, the strength of association is smaller in the case of fungi, compared to bacteria, which presents a slightly higher correlation. The correlations between the diameter of the inhibition area and the flavonoids are stronger than in the case of phenols.

Our study had certain limitations. On the one hand, this study was performed on propolis samples only from four regions of Romania, located in the northwestern part of the country and not from all regions. On the other hand, the number of species of microorganisms was relatively small, with only five bacterial strains and five fungi, but those that frequently contaminate cereals were chosen. In future studies, we propose to analyze other propolis samples from all over the country, to determine the antimicrobial effects on other pathogenic strains for cereals, to determine the minimum inhibitory concentration, the minimum bactericidal and fungicidal concentration. We also propose to evaluate the phyto-inhibitory effect on other plant species that can be harmful to cereal crops.

## 4. Materials and Methods

### 4.1. Propolis Samples

Brown propolis samples produced by bees in wooden hives were collected from 16 counties of 4 regions in Romania (Banat, Crișana, Maramureș, and Transylvania). The samples were collected in June–July 2021. Sampling was performed by scraping off the cover and entering the hives with a stainless-steel spatula. The samples were stored at −18 °C in the darkness until analysis. Table 8 shows the county and the relief related to the area where the samples were taken.

Figure 7 shows the positioning of Romania on the European map and the counties from which samples were taken, respectively.

### 4.2. Physico-Chemical Analysis

#### 4.2.1. Water Activity

The water activity (a_w_) was determined at 25 °C with the Aquaspector apparatus AQS-2-TC (Nagy Messsysteme GmbH, Gäufelden, Germany). The instrument was calibrated. The measurements were repeated three times for each sample [58]. 

#### 4.2.2. Water Solubility

The method described by Cano-Chauca et al. (2005) consisted of dissolving 2 g of propolis sample in 20 mL distilled water and centrifugation at 5000 rpm for 5 min. The aliquot (5 mL) was dried at 105 °C in an oven. Solubility was determined as the sample mass obtained after drying [59]. 

#### 4.2.3. Determination of Total Phenolic Content (TPC)

For the determination of Total Phenolic Content (TPC), the Folin–Ciocalteu method was used [60,61]. 0.5 g of propolis and 15 mL of ethanol were homogenized at 500 rpm (30 min), then filtered and stored in the dark. The same amount of Folin–Ciocalteu reagent was added to 500 μL of ethanolic propolis extract. Thereafter, 2 mL of 10% sodium carbonate solution and distilled water were added for a final volume of 50 mL. The absorbance at 765 nm was measured with the spectrophotometer (Lambda 20—Perkin Elmer UV/VIS, Washington, DC, USA) with distilled water as a control. Total phenol content was determined by interpolating the absorbance of the propolis based on a calibration curve constructed with standard gallic acid, with a purity of 98%.

#### 4.2.4. Total Flavonoid Content (TFC)

For the determination of total flavonoid content, 1 g propolis and 25 mL of 95% ethanol were homogenized at 200 rpm (24 h) and then filtered, adjusted to 25 mL with 80% ethanol, and stored in the dark place. Diluted standard solutions (0.5 mL) were mixed separately with 1.5 mL of 95% ethanol, 0.1 mL of AlCl_3_ 10%,0.1 mL of 1M potassium acetate, and 2.8 mL of distilled water. After 30 min in a dark place, the absorbance readings at 415 nm were determined by spectrophotometer (Lambda 20—Perkin Elmer UV/VIS, Waltham, MA, USA). The total flavonoid content was established using a standard curve, with quercetin as the standard. The mean of three readings was used and expressed as mg of quercetin equivalents (QE)/g of propolis [62].

#### 4.2.5. Ferric-Reducing Antioxidant Power (FRAP)

The ferric-reducing antioxidant assay (FRAP) was performed to highlight the reducing power of propolis extracts. The FRAP reagent was obtained from 2.5 mL of 10 mM 2,4,6 tripyridyl-S-triazine solution (TPTZ reagent) with 2.5 mL of 20 mM FeCl_3_·6H_2_O solution in 25 mL of 300 mM acetate buffer (pH 3.6). Thereafter, 1.5 mL of freshly prepared FRAP reagent was mixed with 200 µL of methanolic extracts of propolis (1 g in 7 mL methanol) and incubated at 37 °C for 4 min.

The absorbance was recorded at λ = 593 nm after prior calibration of the spectrophotometer with 200 µL of distilled water instead of the propolis sample. The standard curve was created by reacting ferrous sulfate (151.5–9.5 mg/mL) with the FRAP reagent [63,64].

#### 4.2.6. The Antioxidant Activity of Propolis

The raw propolis samples were macerated and continuously homogenized for 24 h in 70% ethanol solution (1:100 *w/v*), and then evaporated to dryness. A reaction mixture containing 2,2-diphenyl-1-picrylhydrazyl (DPPH) 0.1 mM ethanoic solution and 0.6 mg/mL propolis solution was prepared. The absorbance was measured in a quartz cuvette (1 cm^3^) at λ = 515 nm with a Lambda 20 UV VIS Spectrophotometer (Perkin Elmer UV/VIS, Washington, DC, USA). Absorbance (A) was measured at the initiation of the reaction, then after 10 and 20 min. The antioxidant activity was calculated using the formula [65,66]: %RSA = (ADPPH − Asample)/ADPPH × 100.

### 4.3. The Phyto-Inhibitory Activity of Propolis

The phyto-inhibitory activity is based on the estimation of the germination period (slowing down) of cereal samples with physical-chemical characteristics in standard systems, with and without the controlled addition of propolis. The global sample was made by mixing equal amounts of propolis from each county. To evaluate the phyto-inhibitory activity, aqueous propolis solutions of different concentrations were prepared: 1%, 5%, and 10%. The cereals used for the study were Hexaploid bread wheat (*Triticum aestivum*): moisture 13.8%, hectoliter weight 77.1 kg/hL at 26.5 °C; Maize (*Zea mays* L.): moisture 14.4%, hectoliter weight 73.8 kg/hL at 26.5 °C; Oats (*Avena sativa* L.): moisture of 12.9%, hectoliter weight of 41.1 kg/hL at 26.4 °C; and Barley (*Hordeum vulgare* L.): moisture 14.2%, hectoliter weight 63.7% at 26.6 °C [67,68].

Aqueous propolis extract (APE) was obtained by the method described in our previous studies. The aqueous propolis extract was obtained from 50 g of propolis, finely chopped, using a mortar and pestle, weighed on a Kern ABT120- 5DNM analytical balance (Kern & Sohn GmbH, Balingen, Germany), to which 250 mL of distilled water was added and refluxed for 1 h in a round bottom flask with a condenser. The heterogeneous system was centrifuged (~4500× *g*) in a centrifuge Centra CL2 (Thermo Fisher Scientific Inc., Waltham, MA, USA), coarsely filtered through a vacuum-connected filter (Merck KGaA, Darmstadt, Germany) by centrifugation at ~4000× *g*, filtered through a low porosity surface connected to a vacuum, and boiled at 100 °C until 20% of the initial quantity remained [69,70]. 

APE was introduced into Petri dishes (20 cm^2^) with a layer of hydrophilic wool, using concentrations of 1% (0.01 g/mL), 5% (0.05 g/mL), and 10% (0.1 g/mL), respectively. The studied cereals were introduced into the formed medium, and every other day for 13 days, and statistical evaluations (averages) were performed on 10 sprouted seedlings.

### 4.4. Antimicrobial Activity of Propolis

#### 4.4.1. Cultures of Microorganisms

In order to evaluate the antibacterial activity of propolis extracts, five strains of bacteria were used, selected from the main species found on cereals: *Pseudomonas fluorescens* (ATCC 13525), *Bacillus subtilis* (ATCC 6633), *Bacillus cereus* (ATCC 11788), *Escherichia coli* (ATCC 25922), and *Proteus mirabilis* (ATCC 7002). Antifungal activity was also evaluated using five strains of fungi from the species that contaminate cereals in the field, before harvesting: *Alternaria alternata* (TX 8025), *Cladosporium cladosporioides* (derived from ATCC 16022), *Fusarium oxysporum* (ATCC 48112), *Mucor racemosus* (derived from ATCC 42647), and *Aspergillus niger* (derived from ATCC 16888). All strains used were provided by Thermo Fisher Scientific Inc. (Waltham, MA, USA) and MicroBioLogics Inc. (St. Cloud, MN, USA).

In order to obtain bacterial cultures, 3–5 colonies from each bacterial strain were dispersed in 10 mL of nutrient broth (Mikrobiologie Labor-Technik, Arad, Romania) and were incubated for 18 ± 2 h, at 37 ± 1 °C. To obtain fungal cultures, colonies from each fungal strain dispersed in 10 mL of nutrient broth were incubated for 72 ± 2 h at 25 ± 1 °C. The turbidity of the cell suspension was measured using a McFarland Densitometer (Mettler Toledo, Columbus, OH, USA) and adjusted until the turbidity of the suspension was equivalent to the turbidity of a 0.5 McFarland standard.

#### 4.4.2. Determination of the Antimicrobial Properties of the Aqueous Propolis Extracts–Agar Disk Diffusion Method

To test the antimicrobial properties of the APE, a disk diffusion method was used according to CLSI-recommended procedures [71] by measuring the diameters of the zones of inhibition produced by microbial strains. The diameter of the inhibition zone is a semi-quantitative measure of the antimicrobial activity.

For bacterial strains, Mueller–Hinton agar (Merck KGaA, Darmstadt, Germany) was used as a culture medium, and for the fungal strain, Sabouraud 4% dextrose agar (Merck KGaA, Darmstadt, Germany) was used. The culture medium was put in Petri dishes with a depth of ~4 mm and the surface was inoculated by flooding with 1 mL culture, then spread on the surface. To absorb the inoculum in the agar, the plates were kept for 15 min at 37 °C after inoculation.

From each sample of propolis, 50 μL of APE 0.1 g/mL (obtained as described earlier) were added to ~6 mm filter paper disc. The discs were deposited sterilely on the surface of the culture medium and kept for 120 min at 5 °C. All discs were applied at approximately the same distance from the edge of the plate and from each other. The Petri dishes were incubated for 24 h at 37 °C for bacterial growth and 5 days at 25 ± 1 °C for fungal growth. Discs with 5 µg ciprofloxacin (Bio-Rad, Hercules, CA, USA) were used as a positive control for bacterial growth. The final evaluation of the antimicrobial activity was made by quantifying (in mm) the diameter of the zones of inhibition obtained, using a DIN 862 ABS digital caliper (Fuzhou Conic Industrial Co. Ltd., Fuzhou, China). All tests were performed in triplicate by the same operator and under the same laboratory conditions, the results being expressed as the average of the three tests (in mm without decimals).

### 4.5. Statistical Analysis

Analysis of variance (ANOVA) was used to test the main and interaction effects of the strain type (bacterial and fungal) and the geographical regions of propolis on the diameter of the inhibition zone. Pearson correlation coefficient measures the strength and the direction of a linear relationship between the diameter of the inhibition zone and the flavonoid and phenol content of propolis samples for the microorganisms studied [72].

## 5. Conclusion

The study characterized propolis samples collected from every county of Transylvania, Romania, by determining the main chemical components. The samples with the highest content in phenols and flavonoids are those from Caraș Severin, Brașov, and Bistrita-Năsăud counties. The antioxidant capacity of propolis samples depends on their total phenolic and flavonoid content.

At the same time, the study highlights the phyto-inhibitory effect of propolis samples on different cereals: hexaploid bread wheat (*Triticum aestivum*), maize (*Zea mays* L.), oats (*Avena sativa* L.), and barley (*Hordeum vulgare* L.). It increases with the increase in propolis concentration.

The propolis samples present antimicrobial activity against all studied bacterial and fungal strains. The most sensitive strangers were *P. fluorescens* among the bacterial strains frequently found on plants, and *F. oxysporum* among the fungal species, respectively. 

Correlations were found between antioxidant activity and flavonoid and phenol content of propolis samples. Satisfactory associations were found between the diameter of the inhibition zone and the content of flavonoids and phenols in propolis for almost all microbial strains. A correlation between the geographical origin of Romanian propolis, the type of strain (bacteria and fungi), and the diameter of the inhibition zone was observed in the study.

The results of the study suggest that aqueous extracts of propolis can be used as an antimicrobial agent for cereals, however, on the other hand, propolis also has a phyto-inhibitory effect on these cultures.

## Figures and Tables

**Figure 1 antibiotics-12-01015-f001:**
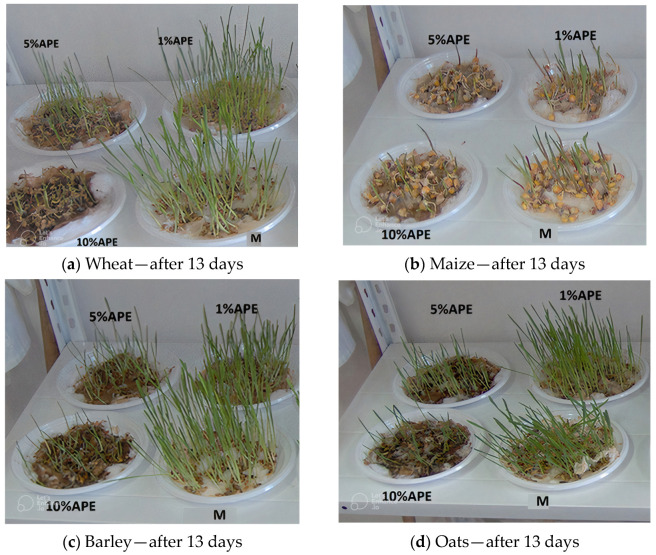
Cereal samples ((**a**)—Wheat, (**b**)—Maize, (**c**)—Barley, (**d**)—Oats) treated with different concentrations of aqueous propolis solutions (APE) after 13 days, M—control sample.

**Figure 2 antibiotics-12-01015-f002:**
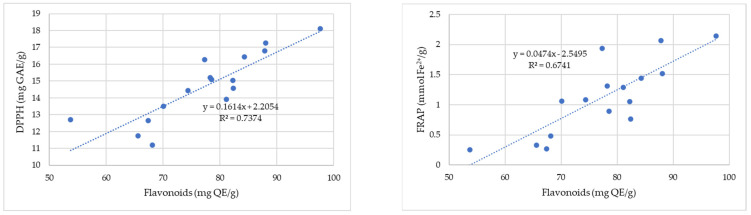
Correlation between antioxidant capacity determined by the DPPH (gallic acid equivalent (GAE)/g) and FRAP (mmol Fe^2+^/100 g) methods and total flavonoid content of the propolis samples.

**Figure 3 antibiotics-12-01015-f003:**
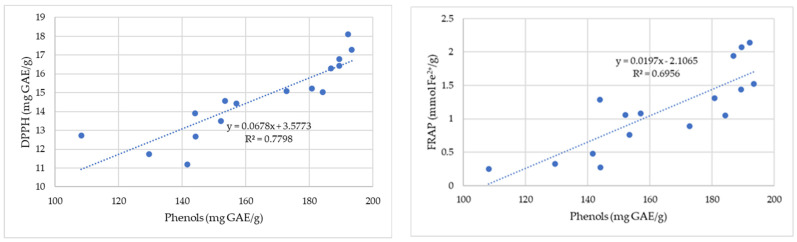
Correlation between antioxidant capacity determined by the DPPH (gallic acid equivalent (GAE)/g) and FRAP (mmol Fe^2+^/100 g) methods and total phenolic content for the propolis samples.

**Figure 4 antibiotics-12-01015-f004:**
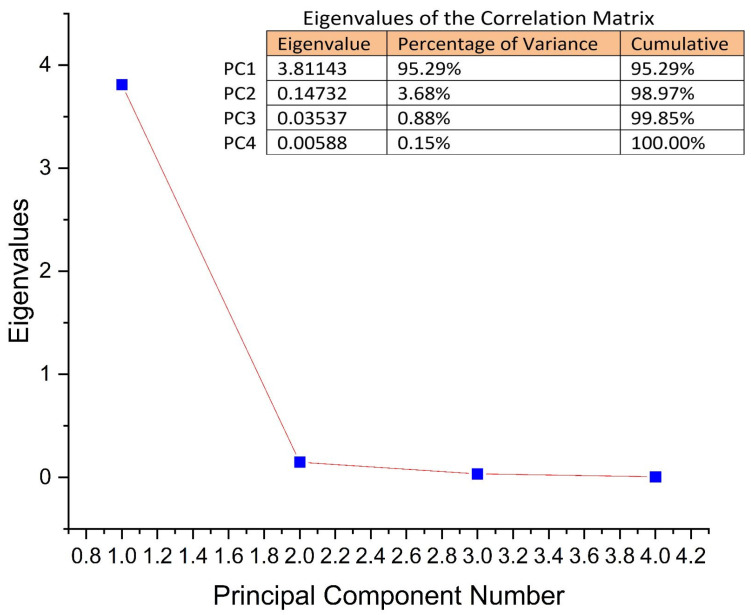
The explained PCs of the total variance and eigenvalues for each of the PCs.

**Figure 5 antibiotics-12-01015-f005:**
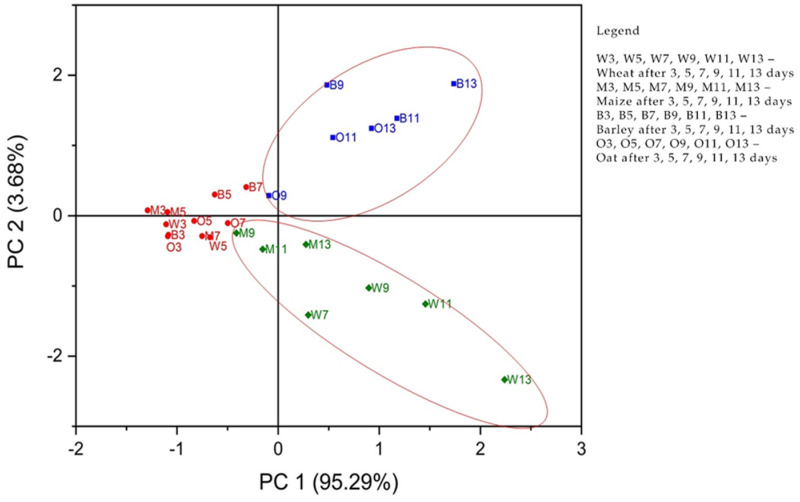
Scores plot for PCA analysis of the four samples of cereals according to their responses to propolis concentrations C(1%), C(5%), and C(10%) control, respectively.

**Figure 6 antibiotics-12-01015-f006:**
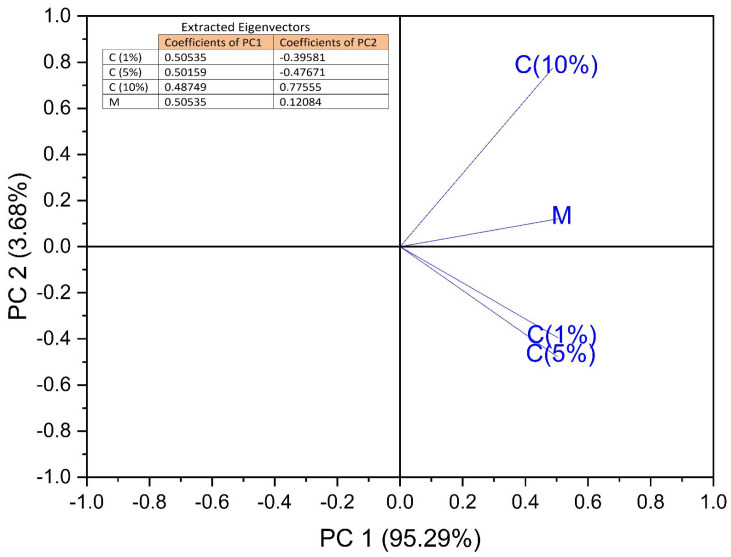
Loading plot for the PCA analysis, contribution of variables, propolis concentrations C (1%), C (5%), and C (10%), Mcontrol, respectively.

**Figure 7 antibiotics-12-01015-f007:**
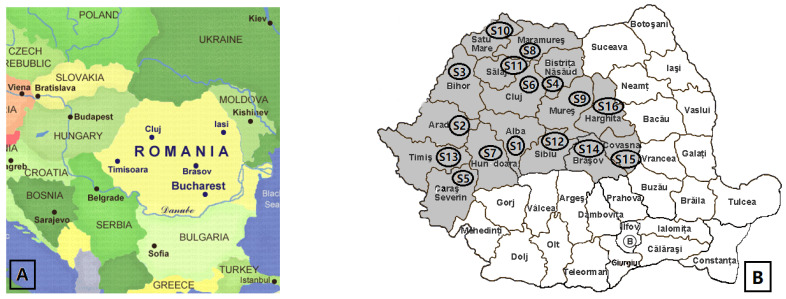
Location of Romania in Europe (**A**); Map of Romania–Transylvania with propolis sampling counties (**B**).

**Table 1 antibiotics-12-01015-t001:** Characterization of brown propolis samples.

Sample No.	Water Activity (a_w_)	Water Solubility (%)	Phenols (mg GAE/g)	Flavonoids (mg QE/g)	FRAP ^1^ (mmol Fe^2+^/g)	DPPH ^2^ (mg GAE/g)
S1	0.71 ± 0.22	9.12 ± 0.27	189.4 ± 5.82	84.31 ± 0.09	1.44 ± 0.31	16.44 ± 0.2
S2	0.74 ± 0.14	12.07 ± 0.31	180.8 ± 4.54	78.26 ± 0.07	1.31 ± 0.01	15.21 ± 0.3
S3	0.69 ± 0.15	8.98 ± 0.66	172.9 ± 3.25	78.55 ± 0.08	0.89 ± 0.02	15.08 ± 0.2
S4	0.73 ± 0.13	14.23 ± 0.49	189.5 ± 4.83	87.84 ± 0.11	2.07 ± 0.08	16.79 ± 0.1
S5	0.74 ± 0.12	11.35 ± 0.57	193.4 ± 7.22	88.06 ± 0.08	1.52 ± 0.02	17.27 ± 0.4
S6	0.71 ± 0.08	13.10 ± 0.72	129.6 ± 3.58	65.59 ± 0.09	0.33 ± 0.09	11.75 ± 0.1
S7	0.73 ± 0.14	9.86 ± 0.14	184.3 ± 6.04	82.27 ± 0.25	1.05 ± 0.03	15.04 ± 0.6
S8	0.62 ± 0.17	8.74 ± 0.50	152.2 ± 6.80	70.10 ± 0.16	1.06 ± 0.06	13.50 ± 0.3
S9	0.67 ± 0.15	13.01 ± 0.34	157.1 ± 5.57	74.35 ± 0.36	1.08 ± 0.14	14.43 ± 0.2
S10	0.66 ± 0.08	10.56 ± 0.68	186.9 ± 6.88	77.33 ± 0.21	1.94 ± 0.07	16.28 ± 0.2
S11	0.72 ± 0.11	15.61 ± 0.63	144.2 ± 5.51	67.41 ± 0.14	0.27 ± 0.01	12.66 ± 0.1
S12	0.65 ± 0.07	11.00 ± 0.19	153.5 ± 4.78	82.38 ± 0.27	0.76 ± 0.05	14.57 ± 0.4
S13	0.74 ± 0.09	9.52 ± 0.27	144.0 ± 2.09	81.09 ± 0.98	1.29 ± 0.06	13.92 ± 0.5
S14	0.69 ± 0.11	15.23 ± 0.71	192.2 ± 1.18	97.65 ± 0.73	2.14 ± 0.09	18.11 ± 0.1
S15	0.68 ± 0.12	13.09 ± 0.28	108.2 ± 4.78	53.72 ± 0.12	0.25 ± 0.03	12.72 ± 0.6
S16	0.70 ± 0.16	14.55 ± 0.67	141.7 ± 2.07	68.15 ± 0.42	0.48 ± 0.04	11.20 ± 0.2

^1^ FRAP value—expressed as mM conc. of Fe^2+^, obtained from a dilution of ferrous sulphate solution having an equivalent antioxidant capacity. ^2^ DPPH value—expressed as gallic acid equivalent (GAE) having an equivalent antiradical capacity.

**Table 2 antibiotics-12-01015-t002:** Average plumule growth lengths in mm for cereal samples treated with propolis for the 16 counties.

Day	1% APE ^1^	5% APE	10% APE	M ^2^
	Wheat			
3	19 ± 1	8 ± 1	2 ± 0.5	24 ± 6
5	36 ± 2	24 ± 1	11 ± 1.5	39 ± 4
7	79 ± 3	58 ± 4	23 ± 3	82 ± 5
9	95 ± 5	81 ± 4	41 ± 8	101 ± 7
11	115 ± 7	102 ± 6	52 ± 7	123 ± 9
13	150 ± 8	137 ± 7	64 ± 8	145 ± 17
	Maize			
3	10 ± 2	3 ± 1	0 ± 0	15 ± 1
5	21 ± 3	6 ± 2	4 ± 1	23 ± 1
7	34 ± 6	20 ± 3	9 ± 3	37 ± 3
9	44 ± 9	31 ± 5	16 ± 3	54 ± 6
11	53 ± 7	42 ± 6	20 ± 5	65 ± 7
13	65 ± 11	58 ± 9	30 ± 8	83 ± 7
	Barley			
3	19 ± 1	12 ± 1	2 ± 0	23 ± 4
5	34 ± 2	23 ± 3	17 ± 1	40 ± 6
7	45 ± 5	31 ± 2	24 ± 1	55 ± 9
9	70 ± 4	44 ± 3	52 ± 3	90 ± 7
11	92 ± 8	82 ± 6	69 ± 5	101 ± 8
13	107 ± 13	96 ± 7	83 ± 6	132 ± 11
	Oat			
3	21 ± 3	9 ± 2	1 ± 0	25 ± 6
5	30 ± 5	18 ± 5	10 ± 1	30 ± 3
7	43 ± 5	28 ± 4	17 ± 2	44 ± 5
9	50 ± 6	42 ± 4	28 ± 2	65 ± 9
11	70 ± 5	51 ± 6	45 ± 4	102 ± 11
13	81 ± 8	62 ± 9	53 ± 7	123 ± 13

^1^ APE—aqueous propolis extract. ^2^ M—control sample.

**Table 3 antibiotics-12-01015-t003:** Inhibition diameter area (mm) produced by the aqueous propolis extracts (0.1 g/mL) on the bacterial strains.

Sample No.	Strain				
	*P. fluorescens*	*B. subtilis*	*B. cereus*	*E. coli*	*P. mirabilis*
S1	32	28	25	32	29
S2	28	26	27	24	21
S3	30	25	26	21	24
S4	28	24	27	26	28
S5	29	28	28	24	23
S6	32	27	29	28	31
S7	31	29	28	27	29
S8	29	26	27	23	25
S9	27	24	26	25	27
S10	30	27	25	29	26
S11	30	25	27	22	22
S12	28	27	26	29	21
S13	29	26	26	26	27
S14	33	29	29	31	31
S15	27	25	27	18	23
S16	29	23	24	22	19
Ciprofloxacin	24	30	30	29	28

**Table 4 antibiotics-12-01015-t004:** Inhibition diameter area (mm) produced by the aqueous propolis extracts (0.1 g/mL) on the fungal strains.

Sample No.	Strain				
	*A. alternata*	*C. cladosporioides*	*F. oxysporum*	*M. racemosus*	*A. niger*
S1	25	23	28	27	24
S2	16	20	23	22	21
S3	19	23	24	24	19
S4	21	19	25	24	23
S5	20	21	26	21	26
S6	23	26	23	25	17
S7	26	24	22	28	18
S8	22	19	27	23	16
S9	24	22	21	20	22
S10	16	24	25	22	15
S11	18	18	23	25	19
S12	20	23	26	26	18
S13	21	20	22	26	16
S14	25	25	27	23	25
S15	19	17	23	22	15
S16	17	20	24	22	18

**Table 5 antibiotics-12-01015-t005:** Equations and adequacy indicators for the obtained statistical models.

Grain	Equation	σ^2^	σ	r^2^	r
Wheat	y = 13.688 + 9.383·x_1_ − 4.836·x_2_	228.928	15.130	0.896	0.947
Maize	y = 7.022 + 4.038·x_1_ − 2.348·x_2_	48.808	6.466	0.902	0.950
Barley	y = 0.432 + 7.659·x_1_ − 1.875·x_2_	68.590	8.280	0.944	0.971
Oat	y = 7.770 + 5.164·x_1_ − 2.212·x_2_	24.85	4.98	0.958	0.979

The values of the concordance indicators argue for a good capacity to predict statistical models.

**Table 6 antibiotics-12-01015-t006:** Bifactorial variance analysis for samples from Banat, Crișana, Maramureș, and Transylvania regions, Romania.

Dispersion Sum of the Diameters of Inhibition Zones	Quadratic Sum		Degrees of Freedom (ν)	Variance		F_computed_		F_0.05_
	Bacteria	Fungi		Bacteria	Fungi	Bacteria	Fungi	
Between propolis extracts	177.25	273.50	15	11.82	18.23	2.82	3.05	1.84
Between strains	258.35	200.69	4	64.59	50.17	15.43	8.40	2.53
Residual	251.15	358.50	60	4.19	5.98	-		-

**Table 7 antibiotics-12-01015-t007:** Pearson’s correlation coefficients between the diameter of the inhibition zone and the flavonoid and phenol content of propolis samples for the microorganisms studied.

Microbial Strains	Flavonoids	Phenols
*P. fluorescence*	0.41	0.34
*B. subtilis*	0.45	0.47
*B. cereus*	0.49	0.32
*E. coli*	0.68	0.54
*P. mirabilis*	0.34	0.28
*A. alternate*	0.32	0.19
*C. cladosporioides*	0.49	0.38
*F. oxysporum*	0.39	0.44
*M. racemosus*	0.21	0.05
*A. niger*	0.64	0.66

**Table 8 antibiotics-12-01015-t008:** The area and the origin of the propolis samples.

Sample	County of Origin	Landforms
S1	Alba	Mountainous
S2	Arad	Plain
S3	Bihor	Hilly
S4	Bistrița-Năsăud	Mountainous
S5	Caraș-Severin	Hilly
S6	Cluj	Hilly
S7	Hunedoara	Sub-mountainous
S8	Maramureș	Mountainous
S9	Mureș	Hilly
S10	Satu Mare	Hilly
S11	Sălaj	Sub-mountainous
S12	Sibiu	Sub-mountainous
S13	Timiș	Plain
S14	Brașov	Sub-mountainous
S15	Covasna	Mountainous
S16	Harghita	Mountainous

## Data Availability

Not applicable.

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
