# Peer review of "Phyto-Inhibitory and Antimicrobial Activity of Brown Propolis from Romania"

_antibiotics, 2023, doi:10.3390/antibiotics12061015_

Round 1

Reviewer 1 Report

The authors have conducted a study on the phyto-inhibitory and antimicrobial activity of different brown propolis collected from the counties of four regions in Romania. The article is interesting, however, there are problems in the structuring of the article. A better explanation of the M&M part, reorganization of the results and discussion part is desirable. Here are some suggestions for the authors:

Line 24: [...], and Proteus mirabilis.

Lines 25-26: [...], and Aspergillus niger

Lines 28-34: rewrite these lines of the abstract as they do not emphasize the results obtained or the conclusions drawn in the abstract.

Lines 53-55: previously (lines 45-51) the components of propolis are discussed. I recommend that you delete these lines or else rewrite them in order to merge them in the previous paragraph.

Line 66: in vitro and in vivo in italics

Lines 63-72: I recommend restructuring these 3 paragraphs to follow the thread of the importance of propolis in agriculture more fluently.

Line 82 (results): you need to comment in more detail on the results part, as well as add the tables/figures that appear in the discussion. It is also convenient to divide the results section into several sub-sections to facilitate the reader's understanding.

Line 86: Table

Lines 88-91: check the font in the Antibiotics author instructions template.

Lines 107-108- idem

Lines 116-117 (figure 1): add a higher quality figure.

Line 123 (Table 3): add the values of the control

Line 127 (Table 4): idem

Line 131 (Discussion): no detailed discussion of the results obtained versus those of other works/authors. This section only shows and comments on the results obtained by the authors, including tables and figures that should be included in the Results section. Therefore, it is necessary to reorder the results and discussion sections, as well as a detailed discussion of the results obtained compared to those of other authors with the preparation of comparative tables.

Line 283: eliminate the space between - and 18

Line 291 (4.2. Physicochemical analysis): explain in more detail all the sub-sections of 4.2.

Line: 324: (w/v) in italics

Line 346: delete the seedlings point.

Line 392 (conclusions): What are the conclusions of the study, how will your study be continued, are there any limitations to it, and do they suggest that propolis can be applied as a protector against bacterial and fungal diseases in certain cereals?

Line 421 (references): Review the bibliography and add the DOIs to the articles that are available.

The document is perfectly understandable, only a few errors have been detected that need to be corrected.

Author Response

Dear Reviewer 1,
Thank you for your recommendations.
The text has been improved in several sections, according to your recommendations. We checked and improved our English. Attached are the answers to each observation.

Reviewer 2 Report

In this manuscript, the authors evaluated the properties of propolis collected from different regions in Romania. The authors analyzed the physico-chemical properties, phyto-inhibitory effect, and antimicrobial effects of these samples. While these effects are known to be associated with propolis, the authors’ work expanded these effects to multiple cereal plant, bacteria, and fungal species. The authors also used PCA, two-way ANOVA, and Pearson’s correlation to analyze the data. Below are my specific comments:

1.     Line 111-113, given the growths of the control samples were different among cereal plants, the absolute lengths of cereal samples are not comparable.

2.     For the antimicrobial assays, how many replicates did the authors perform? The disk diffusion method is prone to be interfered by specific conditions.

3.     The authors should mention the concentration of propolis used for the antimicrobial assays in the text. Therefore, the readers could compare the antimicrobial effect with the phyto-inhibitory effects.

4.     For the PCA analysis,

a.     It’s unclear to me what are the samples W3, W5, etc.

b.     The goal of PCA analysis is to reduce the independent dimensions of the samples and cluster them based on similarity. What are the dimensions evaluated here?

c.     The PCA results show that the variance was mainly explained by one principal component. What are the coefficients of this component? This suggests to me that one feature of these samples dominates the variance.

5.     Please include how aqueous propolis extract is prepared in the method.

The sentence in line 97 needs a revise.

In table 1, the S7 was duplicated.

Author Response

Dear Reviewer 2

Thank you for your recommendations. The text has been improved in several sections, according to your recommendations. We checked and improved our English. Attached are the answers to each observation.

Round 2

Reviewer 1 Report

Dear Authors,

I would like to congratulate you on the work you have done in revising the manuscript. However, it is necessary that you revise lines 284-310 since the citations appear with all the authors. Also, please unify the references, since in some references the DOIs appear as doi, DOI, others with https,.....

usechatgpt init success

There are minor typos in the English language that are not of major importance.

usechatgpt init success

Author Response

Dear Reviewer 1,

Reviewer 2 Report

Thanks for the revised manuscript. The highlights of this manuscript to me are the extensive examinations of the phyto-inhibitory and antimicrobial effects of propolis. However, I’m still very confused about the statistical analysis part, especially the PCA analysis.

1)    What is the purpose of this analysis here? The only variables here are the concentrations of propolis. How was the degree of correlation between concentration and cereal length evaluated?

2)    Were the data standardized before feeding into the PCA analysis? The values for C(10%) were generally lower than control conditions, which may lead to a bias toward control conditions. The ratios C/M were even lower, which might explain the smaller contribution of C/M variables in the loading graph.

3)    The sentence in lines 174-176 needs a revision. I couldn’t understand it.

4)    It almost feels to me that a Pearson’s correlation analysis or linear regressions are more appropriate here to analyze the correlation between propolis concentration and cereal growth.  

Other comments:

1.     The introduction section between lines 47-59 needs a revision. The information was duplicated and disordered.

2.     Line 127-129, maze and wheat have very different lengths even in the control conditions. If the authors would like to evaluate the effects between cereal seeds, it’s better to use relative length compared to control.

3.     The concentrations of propolis used in the antimicrobial assays are 0.1g/ml, and in the phyto-inhibitory assays are 1-10%. How are these two units converted into each other?

4.     Line 259-265, these two paragraphs should be combined.

5.     The formatting of the manuscript becomes quite confusing with the track changes. There are multiple duplications of figures, letters, and words throughout the text.

Please check the text throughout and correct the duplication due to track change.

Author Response

Dear Reviewer 2,

Thank you for your recommendations. Attached is the article in pdf format (after the second revision).

The text has been improved according to your recommendations. We checked and improved our English.

Below are our responses to your comments.

 Thanks for the revised manuscript. The highlights of this manuscript to me are the extensive examinations of the phyto-inhibitory and antimicrobial effects of propolis. However, I’m still very confused about the statistical analysis part, especially the PCA analysis.

1)    What is the purpose of this analysis here? The only variables here are the concentrations of propolis.

We have reviewed and re-analyzed the entire PCA according to your suggestion.

The concentrations of considering APE - C(1%), C(5%), C(10%), together with the control sample, were used as input data. The analysis was designed to evaluate the phyto-inhibitory effect of the three aqueous propolis solutions, on the four cereal samples after 3, 5, 7, 9, 11 and 13 days.

How was the degree of correlation between concentration and cereal length evaluated?

We also added a first order multiple linear regression model to estimate the relationship between the grains plume growth, as a dependent variable, and two independent variables: time and concentration of propolis extract applied as an inhibitor.

2)    Were the data standardized before feeding into the PCA analysis? The values for C(10%) were generally lower than control conditions, which may lead to a bias toward control conditions. The ratios C/M were even lower, which might explain the smaller contribution of C/M variables in the loading graph.

In order to eliminate discrepancies and possible interpretation errors in the PCA analysis, only propolis concentrations C(1%), C(5%), C(10%) respectively the control was used for each type of grain. Data were standardized before being entered into PCA analysis.

3)    The sentence in lines 174-176 needs a revision. I couldn’t understand it.

Changes have been made according to your suggestion.

4)    It almost feels to me that a Pearson’s correlation analysis or linear regressions are more appropriate here to analyze the correlation between propolis concentration and cereal growth. 

Thank you for the suggestion. We used a first order multiple linear regression model to determine the grains plume growth, as a dependent variable, depending on time and the concentration of propolis extract applied as an inhibitor. The correlation coefficients prove a strong relationship between the investigated variables.

Other comments:

  1. The introduction section between lines 47-59 needs a revision. The information was duplicated and disordered.

Changes have been made according to your suggestion.

  1. Line 127-129, maze and wheat have very different lengths even in the control conditions. If the authors would like to evaluate the effects between cereal seeds, it’s better to use relative length compared to control.

Changes have been made according to your suggestion. All the comparisons between the growth lengths of the grains were made by comparing only with the related sample without inhibitor.

  1. The concentrations of propolis used in the antimicrobial assays are 0.1g/ml, and in the phyto-inhibitory assays are 1-10%. How are these two units converted into each other?

For the phyto-inhibitory activity we used three concentrations of propolis aqueous extract: 1% (0.01 g/mL), 5% (0.05g/mL) and 10% (0.1 g/mL). We specified in the article now.

To determine the antimicrobial effect, we chose to use only the concentration of 10% (0.1 g/mL), concentration usually used for such tests. Besides this, in a previous study of ours we tested two concentrations of APE, both 0.1 g/mL and 0.01 g/mL, on some microorganisms, but we noticed that at the concentration of 0.01 g/mL the diameters of the inhibition zones were very small, some microorganisms being even resistant. [Mihaela Laura Vica, Mirel Glevitzky , Delia Mirela Tit, Tapan Behl, Ramona Cristina Heghedus-Mîndru, Dana Carmen Zaha, Francesca Ursu, Maria Popa, Ioana Glevitzky, Simona Bungau. The antimicrobial activity of honey and propolis extracts from the central region of Romania. Food Bioscience 41 (2021) 101014] So we chose to use only the concentration of 0.1 g/mL in this study. Also, in future studies, we intend to determine the minimum inhibitory concentration starting from this concentration (0.1 g/mL) and using decimal dilutions.

  1. Line 259-265, these two paragraphs should be combined.

Changes have been made according to your suggestion. It was a Track changes error.

  1. The formatting of the manuscript becomes quite confusing with the track changes. There are multiple duplications of figures, letters, and words throughout the text.

Due to the recommendations of the reviewers, there were many changes in the article. We try to send you the final form without Track changes (in pdf format) for an easier reading of the article. See the attachment.
